# Personalized Neurophysiological and Neuropsychological Assessment of Patients with Left and Right Hemispheric Damage in Acute Ischemic Stroke

**DOI:** 10.3390/brainsci12050554

**Published:** 2022-04-26

**Authors:** Anastasia Tynterova, Svetlana Perepelitsa, Arкady Golubev

**Affiliations:** 1Immanuel Kant Baltic Federal University, 14 Alexander Nevsky St., 236041 Kaliningrad, Kaliningrad Oblast, Russia; sveta_perepeliza@mail.ru; 2V. A. Negovsky Research Institute of General Reanimatology, Federal Research and Clinical Center of Intensive Care Medicine and Rehabilitology, 25 Petrovka Str., Build. 2, 107031 Moscow, Moskovskaya Oblast, Russia; arkadygolubev@mail.ru

**Keywords:** ischemic stroke, psycho-emotional disorders, cognitive impairment, coping strategies, evoked potentials

## Abstract

The leading factors of post-stroke disability are motor disorders and cognitive dysfunctions. The aim of the study was to identify and provide a rationale for the variable early cognitive dysfunction in right and left hemispheric damage in patients with acute stroke. The study included 80 patients diagnosed with ischemic stroke. All patients were assessed for cognitive status, depression, fatigue and anxiety. For objectification, the method of evoked potentials (P300) and neuroimaging were used. Our findings revealed distinguishing features of cognitive dysfunction and identified a combination of the most informative markers characteristic of right and left hemispheric damage in patients with acute ischemic stroke. In patients with damage to the left hemisphere, a predominance of dysregulation syndrome (decrease in executive function and attention) was revealed, accompanied by various disorders such as severe anxiety and fatigue. The causes of this cognitive dysfunction may be directly related to stroke (damage to subcortical structures), as well as to a preexisting reduction in higher mental functions associated with age and vascular conditions. Cognitive impairments in patients with lesions of the right hemisphere were characterized by polymorphism and observed in the mental praxis, speech (with predominant semantic component) and abstract thinking domains. They are closely linked to each other and are more related to the lateralization of the lesion and preexisting neurodegeneration than to the localization of the ischemic lesion. The study of P300-evoked potentials is a good tool for confirming cognitive dysfunction. The latent period of the P300 wave is more sensitive to neurodegeneration, while the amplitude factor characterizes vascular pathology to a greater extent. The results of the study provide a rationale for a comprehensive assessment of lateralization, stroke localization, underlying diseases, neurophysiological parameters and identified cognitive impairments when developing a plan of rehabilitation and neuropsychological measures aimed at cognitive and emotional recovery of patients both in the acute phase of ischemic stroke and when selecting further personalized rehabilitation programs.

## 1. Introduction

Stroke is a global public health problem and one of the main causes of disability and social maladaptation worldwide [1]. The modernization of health care and the use of high technologies have contributed to the reduction of stroke mortality to 123 cases per 100,000 people [2,3]. This was largely achieved through using cutting-edge technologies, such as mechanical thrombectomy using stent retrievers. The development of various endovascular techniques has improved the recanalization rate in patients with cerebral artery occlusions and expanded the “therapeutic window”, which determines the future functional independence and quality of life and social adaptation [4,5,6]. However, from 30% to 50% of patients have residual neurological disorders of various severity [7]. Various movement disorders associated with the dysfunction of various elements of the movement regulation system are leading factors of disability in many patients [8,9]. Cognitive disorders are another group of functional abnormalities that are widely present among patients with ischemic stroke and complicate rehabilitation [10]. The issue of early occurrence of cognitive impairment in cerebrovascular accident (CVA) remains relevant and controversial. Lesions caused by ischemia in strategic areas, such as the hippocampus, prefrontal cortex, anterior parts of frontal lobes, medial parts of temporal lobes, optic tuberosity, limbic system structures, basal ganglia and especially angular gyrus, make the main contribution to post-stroke cognitive dysfunction [11,12]. Other etiopathogenetic causes of post-stroke cognitive impairment may include “silent” strokes and chronic cerebral ischemia preceding CVA [13,14,15]; affective range and fatigue [16,17], preexisting neurodegeneration preceding CVA [18,19]; lateralization (left or right hemisphere) of focal lesions [20,21,22,23].

The importance of studying the causes, pathogenesis, neurophysiological and neuroimaging markers of stroke is necessary for timely diagnosis and the correction of neuropsychological disorders, which will affect the duration and quality of life, as well as the rehabilitation prognosis of patients who suffered a cerebrovascular event [24]. Currently, psychometric scales are the main tools for neuropsychological examination. The Montreal Cognitive Assessment Scale (MoCA) has the greatest prognostic value in patients with baseline neurological deficit on the National Institutes of Health Stroke Scale (NIHSS) of more than two points [25,26]. Instrumental and laboratory diagnostic methods are aimed both at determining risk factors for cerebrovascular disease and assessing the contribution of vascular and neurodegenerative factors to post-stroke cognitive dysfunction [27]. The method of event-related potential (ERP), particularly P300, is currently considered a tool for predicting the functional outcome in patients with stroke and choosing further rehabilitation therapy programs [28]. Certain parameters of ERP reflect not only the topographic localization of the affected structures but also the relationship between the neurodegenerative process and cognitive deficit [29].

## 2. Materials and Methods

### 2.1. Study Design, Setting

This prospective cohort study was approved by the Independent Ethical Committee of the Center for Clinical Research of Kant Baltic Federal University, Kaliningrad, Russia (Protocol No. 2 of 27 April 2021) and was conducted from April to December 2021 at the regional vascular center of the City Clinical Emergency Hospital of Kaliningrad, Kaliningrad, Russia. The aim of this study was to identify relevant indicators of various modalities in patients with damage to the right and left hemispheres in the acute period of stroke. The study included 80 patients diagnosed with ischemic stroke. The diagnosis was verified by using the standard assessment methods for suspected CVA (neurological examination, ultrasound Doppler test of the brachiocephalic vessels, brain CT scan, coagulation test, biochemical blood test, complete blood count, electrocardiogram, SARS-CoV-2 PCR). The assessment of baseline ischemic changes in the middle cerebral artery area on the brain CT was performed using ASPECTS (Alberta stroke program early CT score) [25]. Psycho-neurological and neurophysiological (P300 evoked potential) testing was performed on days 2–3 of hospitalization.

### 2.2. Participants

#### 2.2.1. Patients

Depending on the localization of the stroke, the patients were divided into two groups: Group 1 included 40 patients with right hemispheric lesions, of whom 22 (55%) were men and 18 (45%) were women (mean age of the patients was 62 ± 12.8 yrs.); Group 2 included 40 patients with left hemispheric lesions, of whom 17 (42.5%) were men and 23 (57.5%) were women (mean age of the patients was 61 ± 11.8 yrs.).

There were no significant differences in age and sex between the groups (*p* > 0.05). All patients were standardized according to various parameters, such as age, sex, stroke type and localization of the lesion, psycho-emotional and cognitive disorders and comorbidities.

After admission, vital function monitoring was performed on all patients.

#### 2.2.2. Inclusion and Exclusion Criteria

Inclusion criteria in the study were the confirmed diagnosis of unilateral ischemic stroke in the carotid system; signed informed consent; age 40 to 70 years at the time of the study; hospital admission within the first 24 h of the CVA onset; National Institutes of Health Stroke Scale (NIHSS) score ≤15; clear consciousness of patients at the time of the study, ability to answer the questions posed and to fill out questionnaires independently or with the help of a doctor.

Exclusion criteria were history of mental illness or severe cognitive impairment; decompensated comorbidities; ASPECTS score ≤7 for patients with ischemic stroke; vertebrobasilar stroke. The patients with initial neuroimaging manifestations corresponding to ASPECTS >7 points were included because of the need to select a specific group of the relatively homogeneous patient population. On admission, the NIHSS score was measured. The baseline score in Group 1 patients was 5.0 [4.7;8.3], and in Group 2 patients, it was 6.0 [5.2;7.5].

### 2.3. Outcome Measures

#### 2.3.1. Cognitive Function Research

Cognitive function was assessed using the Montreal Cognitive Test (MoCA) subscales and additional tests for praxis. Episodic memory was assessed by the five-word test (the patient recalls a short list of five words from different semantic groups); executive function was assessed by the patient’s performance in a test of verbal fluency and abstraction and alternating trail making test (4 points maximum).

Attention was assessed using tests for the repetition of a list of digits in forward and backward order, sequential subtraction and vigilance (4 points).

Speech function was studied using tests for the repetition of two syntactically complex sentences (2 points) and for verbal fluency (1 point). Abstract thinking was assessed by describing the similarity of two objects (2 points) and orientation in time and space was evaluated using questions about today’s date and place (6 points). To study praxis, tests for kinetic praxis (the fist-rib-palm test—1 point) were used, and ideatory praxis was assessed by the patient’s performance in acting with imaginary objects (3 points).

#### 2.3.2. Assessment of Affective Disorders and Fatigue

The level of affective disorders was assessed using the Hospital Anxiety and Depression Scale (HADS). The scale is a subjective tool and is used for the screening of anxiety and depression in patients. It is easy to use and process (filling out the questionnaire is neither time-consuming nor difficult). The HADS contains 14 questions: 7 about anxiety symptoms and 7 about depression. The test results can be classified into 3 categories: no anxiety or depression (0–8 points), subclinical (8–10 points) and clinical (11 points and above) anxiety/depression. MFI-20 (Multidimensional Fatigue Inventory—20) was used to assess fatigue. MFI-20 allows the overall score of fatigue to be determined (normally, the total number of points should not exceed 20–30), as well as the level of physical and mental fatigue. If the total score on one of the subscales is above 12, a preliminary diagnosis of “fatigue syndrome” is warranted.

#### 2.3.3. The Hachinski Ischemic Scale

The Hachinski Ischemic Scale was used in patients with dementia (less than 20 MoCA points). The scale allows differentiating major types of dementia, such as primary degenerative, vascular or multi-infarct, and mixed type (<4 points—primary degenerative dementia, 4–7 points—intermediate value, >7 points—vascular dementia).

#### 2.3.4. Neuroimaging

The assessment of neuroimaging parameters (localization of stroke, underlying vascular and neurodegenerative disease) was performed using a Discovery CT 750HD 64-slice CT scanner.

#### 2.3.5. Neurophysiological Testing

For neurophysiological examination, the long-latency auditory ERP (P300) was assessed using the “oddball active paradigm” technique. We used auditory stimulation with separate triggers for triggering and averaging rare (significant) stimuli, i.e., tone clicks with a frequency of 2000 Hz, and frequent (insignificant) stimuli, i.e., clicks with a frequency of 1000 Hz. Stimuli of 50 ms duration and 80 dB intensity were binaurally delivered and appeared at a frequency of 2.3 Hz in a pseudorandom sequence with a 30% probability of occurrence for significant and 70% for insignificant stimuli. ERP was recorded from symmetrical areas of the left and right hemispheres of the cerebral cortex, including frontal (F3–A1, F4–A2) and central leads (C3–A1, C4–A2, Fz–A2, and Cz–A1). The Encephalan-131-03 system (“Medikom MTD”, Taganrog, Russia) was used for the amplification and averaging of EP R300. The neurophysiological study included the measurement of the latency and amplitude of the long-latency auditory ERP (P300), which is most closely related to cognitive processes of perception and attention. The P300 wave is a positive deflection wave (10–20 μV) with a latency of about 300–400 ms, appearing after the presentation of the stimulus, and it has a wide topographic representation with insignificant interhemispheric asymmetry. The latency of P300 reflects the time necessary to estimate the task; the amplitude shows the level of attention (the working memory connected with the current task) [26].

### 2.4. Statistical Analysis

Statistical analysis was performed using SPSS 23 (Statistical Package for the Social Sciences 23) software. The samples were checked for normal distribution using the Kolmogorov–Smirnov test with Lillefors correction. The arithmetic mean (M) and standard deviation (SD) were used for presenting normally distributed data. Student’s *t*-test was used to compare unrelated samples. Qualitative variables were reported as proportions (expressed in percentages). Qualitative variables were compared using the χ2 criterion or Fisher’s exact test. Differences were considered statistically significant at *p* ≤ 0.05.

## 3. Results

### 3.1. Types of Ischemic Stroke, Comorbidities and Neuroimaging Parameters

At the time of the study, the status of all patients who received standard treatment was stable. No hemodynamic or respiratory disorders were detected. Based on the clinical presentation and the results of instrumental examinations, the following types of ischemic stroke, neuroimaging parameters and comorbidities were diagnosed in the patients (see Table 1).

Atherothrombotic stroke was the most common, occurring in 40% of cases in Group 1 and in 50% of patients in Group 2. Lacunar and cardioembolic strokes are the next most frequent ones. There were no significant differences in stroke type between the groups (*p* > 0.05). The neuroimaging methods showed that in Group 1 patients, the ischemic lesion was most often located in the parietal cortex. Lesions localized in the basal ganglia were most typical for Group 2 patients, with 35% prevalence.

After the analysis of other neuroimaging signs, we observed a significant predominance of cortical atrophy (*p* = 0.045) and CT signs of recurrent stroke (*p* = 0.025) in patients of the first group compared with the second group. The predominance of leukoaraiosis was significant in Group 2 patients compared to Group 1 (*p* = 0.00001). Hydrocephalus was an unfavorable underlying condition found in 50% of patients in both Groups 1 and 2.

### 3.2. Cognitive and Psychoemotional Disorders

In order to determine the main domains of cognitive dysfunction, the MoCA questionnaire and ideational and kinetic praxis tests were used [30]. All patients included in the study were found to have cognitive dysfunction. The cumulative score of cognitive decline in Group 1 patients, was 20.3 ± 5.8 points, and in Group 2 patients, it was 19.6 ± 4.9 points (*p* = 0.561).

According to the assessment results (Figure 1), Group 1 patients, as compared with Group 2, had significantly lower scores for ideator praxis, abstract thinking and verbal function (*p* = 0.0001). Whereas Group 2 patients, compared with Group 1, showed clinically significant dysfunction of executive capacity and attention (*p* = 0.00025).

Patients’ affective range and fatigue were assessed using the Hospital Anxiety and Depression Scale (HADS) and the MFI-20 Scale. The data analysis indicated depression to be a clinically relevant factor with significantly higher prevalence in Group 1 patients (10.7 ± 1.7 points, “subclinical depression”) compared to Group 2 (8.1 ± 1.5 points) (*p* = 0.00001). The anxiety levels in both groups were higher than 11 points and corresponded to “clinically severe anxiety.” Fatigue assessment on the MFI-20 scale revealed a high level of general (13.1 ± 2.4 points) and mental (12.1 ± 2.6 points) fatigue in Group 2 patients (Figure 2).

The Hachinsky scale score in Group 1 was 6.5 ± 2.6, indicating a likely mixed (vascular and atrophic) cognitive dysfunction. In Group 2, the mean score was 8.4 ± 3.1, indicating a high probability of vascular etiology of cognitive impairment (*p* = 0.0005).

### 3.3. Study of Cognitive Evoked Potentials

The results of the neurophysiological study of P300 wave using the “oddball active paradigm” technique are presented in Table 2.

In both groups, there was a decrease in the amplitude and prolongation of P300 wave latency vs. the normal references [31]. Data analysis revealed significant (*p* < 0.05) prolongation of P300 wave latency in all leads (from the affected and intact hemispheres) in Group 1 patients, while Group 2 patients demonstrated a decrease in P300 amplitude mainly in left hemisphere leads (Fz–A1, C3, F3). The study revealed distinctive features of psycho-emotional and motivational status and cognitive dysfunction related to the localization of ischemic stroke. Based on the analysis of the clinical and instrumental data obtained, as well as testing patients using specialized scales, the most informative (relevant) markers for each group of patients with ischemic stroke were identified (Table 3).

## 4. Discussion

The acute phase of anterior and middle cerebral artery ischemic stroke in both groups is characterized by moderate cognitive impairment [32] in 50% of patients, as well as early dementia in 15% of them. The results obtained are consistent with the findings of other studies showing that dementia developed in up to 40% of those who suffered CVA [33,34], while moderate cognitive impairment was seen in up to 71% of patients post-stroke [35]. Distinguishing features of cognitive dysfunction in relation to stroke localization were identified. A cognitive deficit in patients with right hemispheric lesions was seen in the ideational praxis, speech and abstract thinking domains. Most frequently, the ischemic stroke lesions in this group of patients were localized in the parietal cortex. This explains the development of early disorders in the praxis domain, as this localization promotes apraxia manifestations [36,37]. Speech impairment in patients in this group associated with the semantic deficit is associated to a greater extent with lateralization of the lesion. Patients with right hemispheric brain damage most often experience difficulties with speech tasks related to lexical processing [38].

An impaired abstract thinking domain correlates with the semantic deficit and is related both to the localization of the ischemic lesion and to the lateralization of the process. The results of the few studies devoted to the neuroanatomy of abstract-logical thinking have shown the connection of this cognitive domain with the parieto-temporal junction and the medial prefrontal cortex of both hemispheres, the left anterior temporal lobe and the right anterior cingulate gyrus [39,40]. Other important predictors of higher cortical impairment in patients with ischemic stroke include cerebral atrophy and sequelae of the previous CVA confirmed by neuroimaging [41,42]. Adverse psycho-emotional status, in particular depression and anxiety, on the one hand, are predictors of early manifestation of cognitive dysfunction [43,44] and on the other hand, may be secondary to cognitive deficit in stroke patients, although this topic remains controversial [45]. Subcortical localization of ischemic stroke lesions in the left hemisphere explains the development of such post-stroke cognitive disorders as decreased executive function and attention. The subcortical mechanism of these cognitive impairments is due to the dissociation of the frontal lobes and subcortical basal ganglia with the underlying damage to the deep white matter of the brain [46]. Diffuse white matter changes in the cerebral hemispheres (leukoaraiosis) are the most common pathogenetic mechanisms of post-stroke cognitive impairment [47]. Existing cerebral damage increases the prevalence of cognitive dysfunction in the first stroke by up to 10% and in recurrent stroke by up to 30% [48]. Cognitive decline in attention and executive function accounts for the high frequency of mental fatigue in patients in this group [49].

In patients with left-sided brain damage, the decrease in P300 amplitude detected in the study of evoked potentials correlates with the severity of cognitive deficit and is confirmed by neuropsychological examination data. This category of patients is characterized by decreased activity in the executive domain and attention level. The symmetrical increase in the latency of the P300 peak in all leads in patients with right hemispheric stroke is not due to the lateralization of cognitive disorders but to the preexisting neurodegenerative conditions, which is confirmed by CT scan data and Hachinsky scale assessment [50,51]. Moreover, increased P300 latency in the group of patients with right-sided ischemic stroke is associated with a high level of depression. Subjects with depression have a decreased concentration, which leads to delayed processing time of the auditory stimulus and is manifested by P300 latency [52]. However, despite numerous studies, the clinical significance of the P300 as a marker of psycho-emotional disorders remains controversial [53].

Limitations in the study included the difficulty of performing MRI on patients with stroke due to the examination protocol according to the protocol (Protocol for patient’s management. Stroke) and the limited possibility of including patients with moderate and severe aphasias (motor and sensory aphasia) in the study due to the difficulties of conducting neuropsychiatric and neurophysiological testing.

The main goal of the comprehensive examination of patients with CVA is to identify neurological deficits and plan early rehabilitation therapy in order to reduce cognitive impairment and restore emotional status. Cognitive neurorehabilitation is a standard component of rehabilitation programs for patients with acute focal brain injury. This therapeutic approach is aimed at improving cognitive functions, such as attention, memory, executive function, perception and praxis. Currently, classical methods of neuropsychological correction and current techniques using the latest computer technologies are used. Today, the use of virtual reality technologies with interactive virtual scenarios that allow the integration of technologies and human senses is very promising. These technologies are aimed at compensating motor and cognitive deficits. The possibility of training artificial neural networks and other machine learning algorithms to create personalized rehabilitation programs for patients with post-stroke cognitive impairment is rather fascinating. The development of individual rehabilitation trajectories is based on the assessment of relevant markers of damage to the right and left hemispheres, such as the prevalence of the cognitive domain, affective disorders and fatigue and preexisting comorbidities. Thus, in the group of patients with ischemic stroke in the right hemisphere, neuropsychological correction should be focused on correcting praxis, speech and depression with an emphasis on the localization of the stroke lesion and preexisting neurodegeneration. Cognitive training in the group of patients with left-sided ischemic lesions should be aimed at the correction and restoration of executive skills, attention and working memory with focus on mental and general fatigue and anxiety. The use of neuroimaging markers is essential when planning current and long-term rehabilitation therapy to determine the localization of the lesion and preexisting conditions and thus create an individualized approach to cognitive rehabilitation not only for the current manifestations but also for predicted cognitive impairment. Moreover, based on the present study, we suggest that the latent period of P300 is associated with the severity of affective and higher mental disorders and can be used as a prognostic marker for depression and cognitive dysfunction.

## 5. Conclusions

Our findings revealed distinguishing features of cognitive dysfunction and identified a combination of the most informative markers characteristic of right and left hemispheric damage in patients with acute ischemic stroke.

In patients with damage to the left hemisphere, a predominance of dysregulation syndrome (decrease in executive function and attention) was revealed, accompanied by various disorders, such as severe anxiety and fatigue. The causes of this cognitive dysfunction may be directly related to stroke (damage to subcortical structures), as well as to a preexisting reduction in higher mental functions associated with age and vascular conditions.

Cognitive impairments in patients with lesions of the right hemisphere were characterized by polymorphism and observed in the mental praxis, speech (with predominant semantic component) and abstract thinking domains. They are closely linked to each other and are more related to the lateralization of the lesion and preexisting neurodegeneration than to the localization of the ischemic lesion. The study of P300-evoked potentials is a good tool for confirming cognitive dysfunction. The latent period of the P300 wave is more sensitive to neurodegeneration, while the amplitude factor characterizes vascular pathology to a greater extent. The results of the study provide a rationale for a comprehensive assessment of lateralization, stroke localization, underlying diseases, neurophysiological parameters and identified cognitive impairments when developing the plan of rehabilitation and neuropsychological measures aimed at cognitive and emotional recovery of patients both in the acute phase of ischemic stroke and when selecting further personalized rehabilitation programs.

## Figures and Tables

**Figure 1 brainsci-12-00554-f001:**
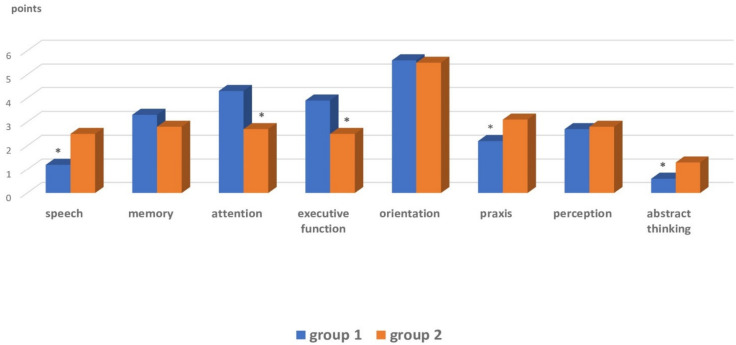
Cognitive function assessment (MoCA scale). Note. * *p* < 0.05—significant intergroup differences.

**Figure 2 brainsci-12-00554-f002:**
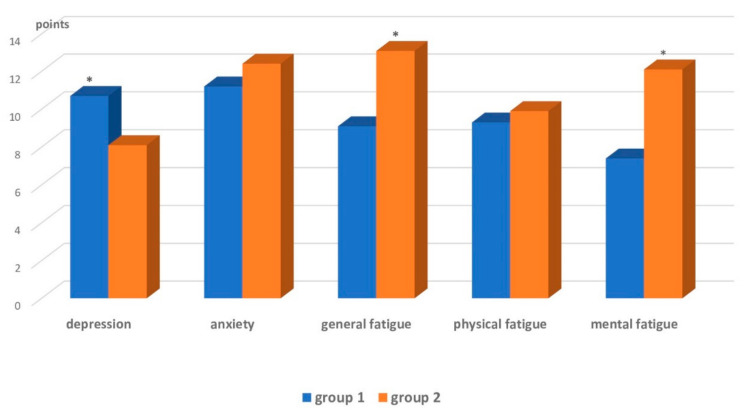
Affective range and fatigue in patients with right and left hemispheric lesions in the acute CVA (HADS, MFI-20). Note: * *p* < 0.05—significant intergroup differences.

**Table 1 brainsci-12-00554-t001:** Types of ischemic stroke and neuroimaging parameters, %.

Type of Ischemic Stroke	Group 1, *n* = 40	Group 2, *n* = 40	*p* Value
Cardioembolic	8 (20%)	7 (17.5%)	0.82
Lacunar	12 (30%)	10 (25%)	0.617
Atherothrombotic	16 (40%)	20 (50%)	0.369
Other	4 (10%)	3 (7.5%)	0.755
	Neuroimaging (localization of the focus)		
Frontal cortex	1 (2.5%)	0%	0.314
Basal ganglia	6 (15%)	14 (35%) *	0.039
Fronto-temporal cortex	4 (10%)	4 (10%)	1.0
Temporal cortex	1 (2.5%)	0%	0.314
Parietal cortex	12 (30%) *	4 (10%)	0.025
Parieto-temporal cortex	2 (5%)	5 (12.5%)	0.211
Parieto-occipital cortex	3 (7.5%)	6 (15%)	0.327
ASPECTS score (points)	9.0 ± 2.7	9.1 ± 2.8	1.0
Neuroimaging (preexisting conditions)			
Cortical atrophy	11 (27.5%) *	4 (10%)	0.045
Leukoaraiosis	2 (5%)	18 (45%) *	0.00001
Hydrocephalus	20 (50%)	20 (50%)	1.0
CT evidence of recurrent stroke	12 (30%) *	4 (10%)	0.025

Note. * *p* < 0.05—significant intergroup differences.

**Table 2 brainsci-12-00554-t002:** Neurophysiological study results (M ± б).

Electrode	Group 1, *n* = 40	Group 2, *n* = 40	*p*-Value	Electrode	Group 1, *n* = 40	Group 2, *n* = 40	*p*-Value
**F3–A1**				**F4–A2**			
Lat P300	476.4 ± 8.7 *	430.4 ± 7.8	0.00001	Lat P300	466.2 ± 8.4 *	436.3 ± 5.6	0.00001
Amp P300	8.0 ± 3.4	6.7 ± 2.1 *	0.043	Amp P300	8.9 ± 3.3	7.6 ± 2.6	0.0539
**C3–A1**				**C4–A2**			
Lat P300	472.2 ± 8.3 *	424.2 ± 7.9	0.00001	Lat P300	485.8 ± 7.9 *	413.4 ± 7.2	0.00001
Amp P300	8.1 ± 2.4	6.8 ± 2.6 *	0.0227	Amp P300	9.0 ± 3.3	7.7 ± 1.9 *	0.0339
**Fz–A1**				**Cz–A2**			
Lat P300	424.3 ± 8.4 *	413.3 ± 7.7	0.00001	Lat P300	455.8 ± 7.6 *	427.7 ± 5.3	0.00001
Amp P300	9.4 ± 3.7	6.3 ± 2.8 *	0.001	Amp P300	9.9 ± 3.9	8.9 ± 3.2	0.2137

Note: * *p* < 0.05—significant intergroup differences. Lat P300–P300 wave latency (ms), Amp P300–P300 wave amplitude (mV).

**Table 3 brainsci-12-00554-t003:** Main relevant parameters of cognitive dysfunction and underlying conditions in patients with acute CVA and right and left hemispheric lesions.

Parameter	Group 1, *n* = 40	Group 2, *n* = 40
Lesion localization	Parietal cortex	Subcortical structures
Neuroimaging markers	Atrophy, signs of recurrent ischemic stroke	Signs of chronic brain ischemia (leukoaraiosis)
Neurophysiological markers	P300 latency prolongation in all leads	Decreased amplitude of P300 in Fz–A1, C3, F3, C4 leads
Cognitive dysfunction	Praxis, speech, abstract thinking	Executive function, attention
Affective range and fatigue	Subclinical depression, clinical anxiety	Increased general and mental fatigue, clinical anxiety
Hachinsky’s scale	Mixed (vascular and atrophic) nature of dementia is likely	Likely vascular dementia

## Data Availability

Not applicable.

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
