# Peer review of "Personalized Neurophysiological and Neuropsychological Assessment of Patients with Left and Right Hemispheric Damage in Acute Ischemic Stroke"

_brainsci, 2022, doi:10.3390/brainsci12050554_

Round 1
Reviewer 1 Report
Thank you for inviting me to review this manuscript, “Personalized neurophysiological and neuropsychological assessment of patients with left- and right hemispheric damage in acute ischemic stroke”. As narrated by the authors, strokes tend to trigger intransigent and debilitating disabilities. Thus, employing an idiographic approach to capture cognitive, emotional, and brain regions affected by stroke would help chart algorithms relevant for the severity of the post-stroke disability, pathways for rehabilitation, and as well identifying prognostic indicators. The added value of this manuscript is that the authors have brought forth the role of lateralisation. Some comments for the authors to consider are detailed below. They may look extensive, but encourage the authors to consider them to improve the presentation of this interesting manuscript.
MAJOR ISSUE
After the last paragraph on line 2, line 67, it would be advantageous for the authors to restate the aims of this study.
The strength of this manuscript is that the authors have employed various outcome measures, including Montreal Cognitive Assessment (MoCA), Hospital Anxiety and Depression Scale (HADS), MFI–20 (Multidimensional Fatigue Inventory - 20), Hachinski Ischemic Score for identification of preexisting neurodegenerative disease, "Fist-rib/ palm-fist” test assessing the kinetic praxis, Test of Ideational Praxis (TIP). This is impressive. However, the authors need to acknowledge the progenitors of these measures by citing relevant literature. Also, for us international readers, we will be interested to know the validity and reliability of these measures for the local population. Were they given in Russian? Please attend to this issue
The authors should make concerted efforts to include all the relevant information that is required to have comprehensive “Materials and Methods”. Please try to adhere to the structure/content of the method as recommended by STROBE (STrengthening the Reporting of OBservational studies in Epidemiology.) that can be found here: https://www.strobe-statement.org/checklists/ . For example, the sample size needs to be elucidated, catchment areas, etc.
The authors have included too many Bullets points in the manuscript. It would be better if some of them are narrated as sentences and they should flow with the previous and subsequent statements.
The cognitive status was measured by Montreal Cognitive Assessment (MoCA). In neuropsychological parlance, MOCA is a bedside test and is only capable to solicit the presence of cognitive decline. Therefore, the semantics should change throughout the manuscript when discussions of cognition arise. While we were not informed in the method section, the authors scored MoCA using different subscales. This is acceptable, but should be explained in the method section. A similar feat would be required for the subscale employed for fatigue- MFI–20 (‘general fatigue’, ‘physical fatigue’, and ‘mental fatigue’). Related to this, in Table 3, these loaded terms “Psychoemotional and motivational status” has heuristic value. But I will simplify them. Since the measure was HADS and MFI–20, I will simply state “affective range and fatigue” or “mood symptoms and fatigue”. This emphasis should be addressed throughout the manuscript.
MINOR ISSUE
Please provide the citation/s for this statement (“Stroke is a global public health problem and one of the main causes of disability and social maladaptation worldwide”).
Please provide the citation/s for this statement (“…..correction of neuropsychological disorders, which will affect the duration and quality of life, as well as the rehabilitation prognosis of patients who suffered a cerebrovascular event”)
In Table 2, too many abbreviations have cropped up. The tables and Figures should be self-descriptive and therefore all abbreviations and acronyms should be defined within the parameter of the tables and Figures.
Reviewer 2 Report
It's an honor for me to review the manuscript number "brainsci-1658993" titled: "Personalized neurophysiological and neuropsychological assessment of patients with left and right hemispheric damage in acute ischemic stroke" for the journal "Brain Sciences".
The authors presented a very interesting original article where the authors collected 80 patients diagnosed with ischemic stroke, which were assessed the cognitive status, depression, fatigue and anxiety. For objectification, the method of evoked potentials (P300) and neuroimaging were used.
The aim of the study was to identify and provide a rationale for the variable early cognitive dysfunction in right and left hemispheric damage in patients with acute stroke. The identified distinctive features of cognitive dysfunction for right and left hemisphere lesions in patients with acute ischemic stroke demonstrate the feasibility of a comprehensive assessment of the most informative parameters in the development of rehabilitation measures aimed at restoring the patients' cognitive-emotional resources
The study is very interesting and well conceived, since it studies an extremely important field for the recovery of the patient's full functionality, especially in social life.
A few comments in order to increase the value of yur work:
General comment:
- Minor english revision needed, there aresome typos and some run-on sentences, sometimes certain sentences cut off the momentum of reading, which could possibly be unpleasant for some readers.
Abstract: can you better explain which is the conclusion of your work?
Introduction: nothing to concern
M&M: why did you exclude patients with an ASPECT<7? this can be a selection bias.
Did you perform just CT for neuroimaging evaluation? this can be a great limit in evaluation of small subcortical lesions and chronic brain ischemia.
Results: clearly presented.
Statistical analysis is generally simple. Could you implement this aspect, in order to value your conclusions?
Discussion: since you analyzed substiantially patients with small core stroke (considering ASPECTs) and with a relatively low NIHSS score (<15), could you could you complement the discussion with recent results from endovascular therapy studies on intracranial occlusions with low NIHSS (eg: - Alexandre AM, Valente I, Frisullo G, Morosetti R, Genovese D, Bartolo A, Gigli R, Rollo C, Scarcia L, Carosi F, Fortunato G, D'Argento F, Calabresi P, Della Marca G, Pedicelli A, Broccolini A. Mechanical thrombectomy in patients with stroke due to large vessel occlusion in the anterior circulation and low baseline NIHSS score. J Integr Neurosci. 2021 Sep 30;20(3):645-650. doi: 10.31083/j.jin2003068. PMID: 34645097.
- Haussen DC, Lima FO, Bouslama M, Grossberg JA, Silva GS, Lev MH, Furie K, Koroshetz W, Frankel MR, Nogueira RG. Thrombectomy versus medical management for large vessel occlusion strokes with minimal symptoms: an analysis from STOPStroke and GESTOR cohorts. J Neurointerv Surg. 2018 Apr;10(4):325-329. doi: 10.1136/neurintsurg-2017-013243. Epub 2017 Aug 2. PMID: 28768820.
- Sarraj A, Hassan A, Savitz SI, Grotta JC, Cai C, Parsha KN, Farrell CM, Imam B, Sitton CW, Reddy ST, Kamal H, Goyal N, Elijovich L, Reishus K, Krishnan R, Sangha N, Wu A, Costa R, Malik R, Mir O, Hasan R, Snodgrass LM, Requena M, Graybeal D, Abraham M, Chen M, McCullough LD, Ribo M. Endovascular Thrombectomy for Mild Strokes: How Low Should We Go? Stroke. 2018 Oct;49(10):2398-2405. doi: 10.1161/STROKEAHA.118.022114. PMID: 30355094; PMCID: PMC6209123.)
In your discussion, i would like you to try to focus more on the major findings of your study. For example in the two last paragraphs please try to express the importance of your work.
At the end of discussion section please try to evaluate which are the major limitation of your work: small groups of patients, incomplete neuroimaging evaluation etc.
I think your conclusion lacks content, please try to summarize your study so that readers who read only the conclusion understand your whole work.
At the end, I hope that the authors will not blame me too much for my remarks, my only objective is to help to increase the overall level of the manuscript, I wish good luck to all the authors.
